# TadA orthologs enable both cytosine and adenine editing of base editors

Shuqian Zhang[1,2,9], Bo Yuan [3,9], Jixin Cao[4], Liting Song[4], Jinlong Chen[1], Jiayi Qiu[1], Zilong Qiu [3,5,6], Xing-Ming Zhao [4,7,8], Jingqi Chen [4,7,8] & Tian-Lin Cheng [1] ✉

Cytidine and adenosine deaminases are required for cytosine and adenine editing of base editors respectively, and no single deaminase could enable concurrent and comparable cytosine and adenine editing. Additionally, distinct properties of cytidine and adenosine deaminases lead to various types of off-target effects, including Cas9-indendepent DNA off-target effects for cytosine base editors (CBEs) and RNA off-target effects particularly severe for adenine base editors (ABEs). Here we demonstrate that 25 TadA orthologs could be engineered to generate functional ABEs, CBEs or ACBEs via single or double mutations, which display minimized Cas9-independent DNA off-target effects and genotoxicity, with orthologs B5ZCW4, Q57LE3, E8WVH3, Q13XZ4 and B3PCY2 as promising candidates for further engineering. Furthermore, RNA off-target effects of TadA ortholog-derived base editors could be further reduced or even eliminated by additional single mutation. Taken together, our work expands the base editing toolkits, and also provides important clues for the potential evolutionary process of deaminases.

Distinct deaminases are required for base editors to induce cytosine or adenine editing, with ABEs containing evolved tRNA-specific adenosine deaminases from *Escherichia coli* (hereafter ecTadA for wild-type version and ecTadA7.10 for evolved version in ABE7.10) to edit adenine(s) and CBEs containing cytidine deaminases to edit cytosine(s) at target genomic regions[1–5]. Cytidine and adenosine deaminases also display distinct properties leading to various types of off-target effects[6–10]. For example, ABEs displayed obvious RNA off-target effects as wild-type ecTadA is an RNA-specific deaminase, while CBEs displayed significant Cas9-independent DNA off-target effects due to intrinsic single-stranded DNA (ssDNA) affinity of

cytidine deaminases[11–17]. As off-target effects severely restricted the applications of base editors, deaminase engineering and screening have been extensively exploited to improve base editing tools[18–24]. Nevertheless, deaminases with optimized properties remain limited, and thus constrained the editing outcomes of base editors. Moreover, as no single deaminase enables comparable cytosine and adenine editing simultaneously, ACBEs require two distinct deaminases fusing together, which leads to a larger size to restrict their utility[25–27].

It has been demonstrated that ABEs could catalyze cytosine deamination within specific sequence contexts, and particular mutations could modulate the editing preference for adenine or cytosine[18,28–31].

[1]Institute for Translational Brain Research, State Key Laboratory of Medical Neurobiology, MOE Frontiers Center for Brain Science, Institute of Pediatrics, National Children's Medical Center, Children's Hospital, Fudan University, Shanghai, China. [2]Department of Pediatrics, Qilu Hospital of Shandong University, Ji'nan 250012, China. [3]Institute of Neuroscience, State Key Laboratory of Neuroscience, CAS Center for Excellence in Brain Science and Intelligence Technology, Chinese Academy of Sciences, Shanghai 200031, China. [4]Institute of Science and Technology for Brain-Inspired Intelligence, Fudan University, Shanghai, China. [5]National Clinical Research Center for Aging and Medicine, Huashan Hopsital, Fudan University, Shanghai 200032, China. [6]Songjiang Hospital, Songjiang Institute, Shanghai Jiao Tong University School of Medicine, Shanghai, China. [7]State Key Laboratory of Medical Neurobiology, Institutes of Brain Science, Fudan University, Shanghai, China. [8]MOE Key Laboratory of Computational Neuroscience and Brain-Inspired Intelligence, and MOE Frontiers Center for Brain Science, Fudan University, Shanghai, China. [9]These authors contributed equally: Shuqian Zhang, and Bo Yuan. ✉e-mail: chengtianlin@fudan.edu.cn

Therefore, TadA deaminases represented promising candidates for the generation of ABEs, CBEs, and ACBEs. However, it is well-known that wild-type ecTadA deaminase could not edit adenines at the DNA level, and more than ten amino acid substitutions, identified by directed protein evolution, were required for ecTadA engineering to generate potent ABEs[2]. In consideration of the complicated ecTadA evolution process, additional TadA orthologs have not been exploited and engineered for functional base editors.

Here, we show that in combined with an internal fusion strategy, functional base editors, including ABEs, CBEs, and ACBEs, are generated with various TadA orthologs containing just one or two amino acid substitutions. Additionally, RNA off-target effects are further minimized by single amino acid substitution. This study provides an alternative and simplified strategy for orthologous TadA screening and engineering, which would be valuable for the expansion and diversification of base editing tools, and also provide insights into the potential evolutionary process of deaminases.

## Results

### Base editing capacity of ecTadA fusing inside nCas9

Directed protein evolution identified A106V and D108N as core substitutions of ecTadA (hereafter ecTadA(VN)) for functional ABEs to achieve adenine editing at the DNA level. However, ecTadA(VN) displayed marginal adenine editing and additional complicated engineering was required to enhance its editing capacity. As ecTadA7.10 insertion inside nCas9 could enhance the editing

capacity and modulate substrate specificity of ABEs[18], we examined whether domain insertion strategy could also enhance the editing capacity of ecTadA(VN)-derived ABEs. In addition to the conventional N-terminus site, ten representative insertion sites inside nCas9, including docking site (DS) 535/583/770/793/801/895/905/919/1029/1249, were selected for ecTadA(VN) insertion and editing capacity was evaluated at a specific endogenous site containing multiple targetable cytosines and adenines (targeting by sgRNA-1) in HEK293T cells. In consistent with previous studies, N-terminal derived N-ecTadA(VN) displayed marginal editing capacity (Fig. 1a, b) while several internal insertion sites enhanced editing capacity and even led to shifted editing scopes, with 1029-ecTadA(VN) displayed the highest adenine editing activity (68% A-to-G conversion frequency at A5) and detectable cytosine editing activity (11% C-to-T conversion frequency at C6) (Fig. 1b). To exclude potential conformational restrictions or steric hindrance on deaminase activity, we further modified ecTadA(VN) domain by adding 5′-NLS and 3′-flexible linker sequences (NLS-ecTadA(VN)-linker, simplified as NL-ecTadA(VN)) to replace original ecTadA(VN) (Fig. 1c). It was revealed that NL-ecTadA(VN) significantly improved the editing capacity of derived ABEs as compared to unmodified ecTadA(VN)-derived ABEs (Fig. 1d), with N-NL-ecTadA(VN) displaying 31% A-to-G conversion frequency at A5. Among all ecTadA(VN)-derived ABEs, 1249-NL-ecTadA(VN) displayed the highest adenine and cytosine editing activity, with 83% A-to-G conversion frequency at A5 and 27% C-to-T conversion frequency at C4 (Fig. 1d). In consideration of

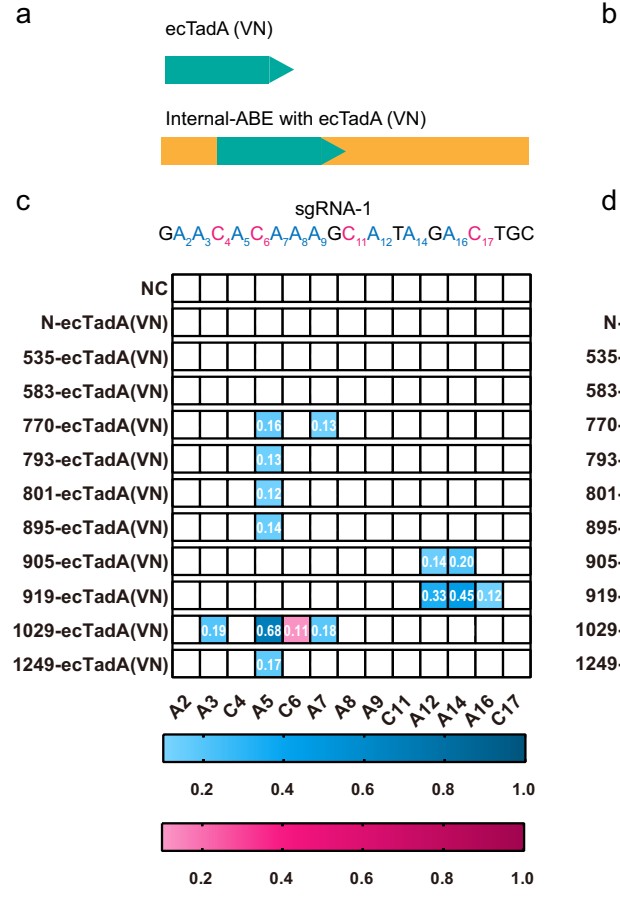

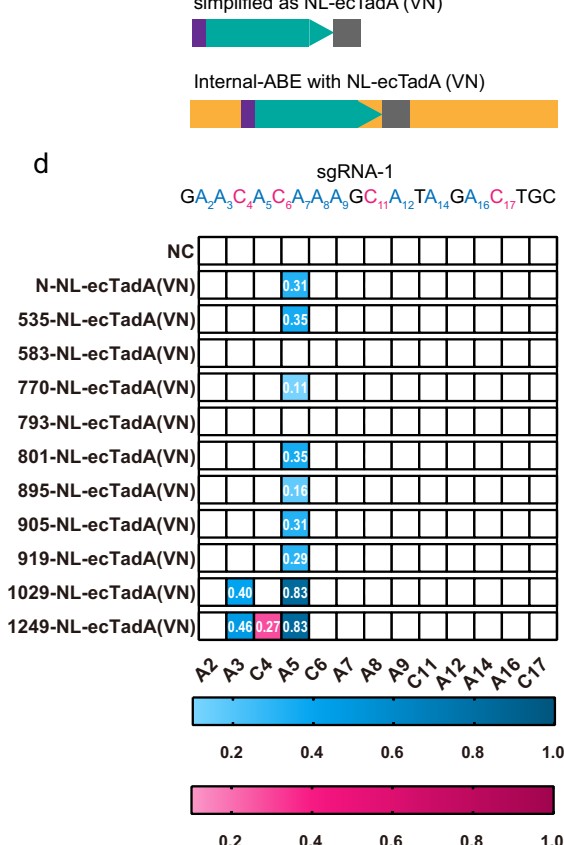

**Fig. 1 | Editing capacity of ecTadA(VN)-derived ABEs. a, b** Illustration of ecTadA(VN) and –derived ABE structures generated via internal insertion strategy. **c, d** The editing frequencies are shown in heatmap format, with adenine editing efficiencies (A-to-G editing) shown in blue and cytosine editing efficiencies (C-to-T editing) in pink gradient color. Engineered ABEs were generated in combination with internal insertion with ecTadA(VN) (**c**) or NLS-ecTadA(VN)-linker (NL-ecTadA(VN)) (**d**). Data shown here represent means of results from $n$ = 3 biologically independent experiments. NC negative control. VN, amino acid substitutions of A106V&D108N. Base editors generated by different fusion strategies were defined as X-deaminase, with X representing fusion sites, such as N for N-terminal and 535 for internal sites after the 535th amino acid. Source data are provided as a Source Data file.

optimal adenine and cytosine editing capacities with 1249-NL-ecTadA(VN), DS1249 insertion site and modified deaminase domain containing 5′-NLS and 3′-flexible linker (hereafter NL-deaminase) were selected for subsequent experiments.

## Screening of TadA orthologs to generate functional ABEs, CBEs, and ACBEs

As ecTadA(VN) fusing inside DS1249 of nCas9 enabled functional ABEs with only A106V and D108N double mutations, we wondered whether other TadA orthologs could also be simply engineered with corresponding double mutations to generate functional ABEs, CBEs, and ACBEs without complicated directed protein evolution. About 1000 TadA orthologs described previously[32] (TadA orthologs used for phylogenetic tree analysis were listed in Supplementary Data 1) were selected for phylogenetic tree analysis with MEGA and classified into six groups (Fig. 2a). Subsequently, 54 TadA orthologs with large evolution distance across all six groups, including four orthologs with known structural information (ecTadA, *Staphylococcus aureus* TadA (Q99W51), *Salmonella choleraesuis* TadA (Q57LE3), and *Streptococcus pyogenes* TadA (Q5XE14)), were chosen for protein engineering and screening (Uniprot ID was used for orthologs nomenclature). Generally, only residues corresponding to A106 and D108 of ecTadA were substituted with V and N in TadA orthologs (detailed protein sequences in Supplementary Data 2), which were further inserted into the DS1249 site to generate potential base editors and assessed at five representative endogenous sites (Supplementary Data 3). Base editors displaying obvious editing activities at 1 or more sites were considered functional. It was revealed that at least 25 TadA orthologs, including ecTadA, could be engineered via N or VN mutations to generate functional base editors, with nine displaying mainly adenine editing activity, nine displaying obvious adenine and cytosine editing activities, and seven displaying mainly cytosine editing activity (Fig. 2b). Though no obvious correlation was observed between substrate specificities and phylogenetic homology for 25 functional TadA orthologs, engineered TadA orthologs of ABEs generally had lower sequence identity to TadAs of CBEs, as compared to those of ACBEs (Fig. 2c).

As TadA was intrinsically an RNA-specific deaminase and RNA off-target effects were particularly severe for ecTadA-derived adenine base editors (ABEs), we evaluated the RNA off-target effects of several representatives TadA ortholog-derived functional base editors. It was revealed that derived functional base editors, regardless of ABEs, ACBEs, or CBEs, all induced detectable A-to-I (average of ~2.5–6.6-fold) but no obvious C-to-U RNA off-target edits as compared to nCas9 (Fig. 3a, b), indicating that the substrate specificity of TadA orthologs at RNA and DNA level was not necessarily correlated and could be reprogrammed separately. Furthermore, as compared to TadA-8e, an evolved ecTadA variant with robust A-to-G editing activity and A-to-I RNA off-target edits (average of ~7.4-fold), all TadA orthologs induced fewer A-to-I RNA off-target edits, with B5ZCW4(VN) inducing the fewest edits (average of ~2.5-fold, Fig. 3a). Additionally, previous studies showed that CBEs induced obvious genotoxicity (manifested as increased accumulation of γH2AX), which was generally attributed to deaminase domain[33,34]. Here we revealed that TadA-8e and ecTadA(VN)-derived ABEs also induced significantly increased γH2AX accumulation (average of 26.1 and 20.8%, respectively) as compared to nCas9 (average of 12.2%), while 6/9 representative TadA ortholog-derived base editors displayed similar γH2AX (average of 13.1, 18.5, 11.2, 10.7, 11.2, 16.7, 12.6, 19.9, and 25.1%, respectively) to nCas9 (Fig. 3c). Therefore, screening and engineering of TadA orthologs could generate functional ABEs, ACBEs, and CBEs with reduced RNA off-target effects and minimized genotoxicity risks.

We subsequently selected six representatives TadA ortholog-derived base editors 1249-NL-B5ZCW4(VN), E8WVH3(VN), Q13XZ4(VN), B3PCY2(VN), Q57LE3(VN), and Q99W51(VN), for editing signature analysis at multiple endogenous sites with high-throughput

sequencing. Additionally, 1249-NL-ecTadA7.10 and -ecTadA(VN) were also included as controls. 1249-NL-ecTadA7.10, with ecTadA7.10 as a significantly improved version, mainly served as ABE and displayed editing scope across A4-A12 with robust editing activity (>60% A-to-G conversion frequency), while 1249-NL-ecTadA(VN) displayed editing scope at A5 with modest editing activity (~40% A-to-G conversion frequency) (Fig. 3d, e). 1249-NL-B5ZCW4(VN) mainly served as ABE, with editing scope across A5-A7 (30–40% A-to-G conversion frequency) (Fig. 3f). 1249-NL-Q57LE3(VN) mainly served as ABE with adenine editing scope at A5 (~40% A-to-G conversion frequency), while 1249-NL-Q99W51(VN) displayed low and comparable adenine and cytosine editing activities at A3, A5, and C5 (10–15% A-to-G and C-to-T conversion frequency) (Fig. 3g, h).1249-NL-E8WVH3(VN) mainly served as ACBE, with adenine and cytosine editing scope across A3-A7 and C4-C6 respectively (~50% A-to-G and ~20% C-to-T conversion frequency) (Fig. 3i). 1249-NL-Q13XZ4(VN) displayed robust cytosine editing activity across C4-C6 and significantly lower but appreciable adenine editing activity at A5, A7 (30–40% C-to-T and ~10% A-to-G conversion frequency) (Fig. 3j). 1249-NL-B3PCY2(VN) mainly served as CBE, with cytosine editing scope across C4-C7 (~40% C-to-T conversion frequency) (Fig. 3k).

We further revealed that the base editors described above displayed diversified sequence preferences for cytosine editing. For orthologous TadA-derived base editors, it was revealed that 1249-NL-E8WVH3(VN) preferred CW (CA/CT > C̲C̲ > CG) and AC, 1249-NL-Q13XZ4(VN) preferred CW (CA/CT > C̲C̲/CG) and WC (AC/TC > C̲C̲/GC) while 1249-NL-B3PCY2(VN) preferred CT/AC and displayed minimal cytosine editing activity at CG/GC contexts (Supplementary Fig. 1).

We further analyzed Cas9-independent DNA off-target effects, which were the major safety risks of cytidine deaminase-derived CBEs, for selected representative base editors using the R-loop assay described previously[19]. It was shown that, at all five examined R-loop sites, TadA ortholog-derived base editors displayed no significant adenine or cytosine editing activities, with the highest adenine or cytosine editing activity at just ~2% (Supplementary Fig. 2a, b).

## Engineering of representative TadA orthologs to minimize RNA off-target effects

As described above, TadA ortholog-derived base editors induced increased A-to-I RNA off-target edits as compared to nCas9, so we wondered whether RNA off-target effects could be minimized through deaminase engineering. It has been reported previously that specific ecTadA mutagenesis, such as F148A (simplified as FA)[13], R153 deletion (simplified as Rdel)[23], or V82G/W (simplified as VG or VW)[12], could significantly reduce or even eliminate RNA off-target edits of ecTadA-derived ABEs. Here we introduced similar mutations into representative 1249-NL-Q13XZ4(VN), -B3PCY2(VN), -E8WVH3(VN), -Q57LE3(VN), and -Q99W51(VN), which displayed obvious A-to-I RNA off-target edits, and evaluated the changes of editing activity and RNA off-target edits at specific sites. It was revealed that Q13XZ4(VN) (FA), B3PCY2(VN) (Rdel), E8WVH3(VN) (FA), and Q57LE3(VN) (Rdel) significantly reduced A-to-I RNA off-target edits at the selected site while maintaining comparable on-target editing activities (Fig. 4a). We further determined the RNA off-target edits by RNA sequencing and confirmed that base editors containing Q13XZ4(VN) (FA), B3PCY2(VN) (Rdel), E8WVH3(VN) (FA), and Q57LE3(VN) (Rdel) displayed similar A-to-I and C-to-U RNA off-target edits to nCas9 (Fig. 4b, c).

## Discussion

It is widely believed that complicated engineering of adenosine deaminases via directed protein evolution was necessary to generate functional ABEs and all available ABEs were derived from ecTadA with more than ten amino acid substitutions[2]. Here we demonstrated that in combined with an internal fusion strategy at specific insertion sites of nCas9, engineering of ecTadA with just two amino acid substitutions at

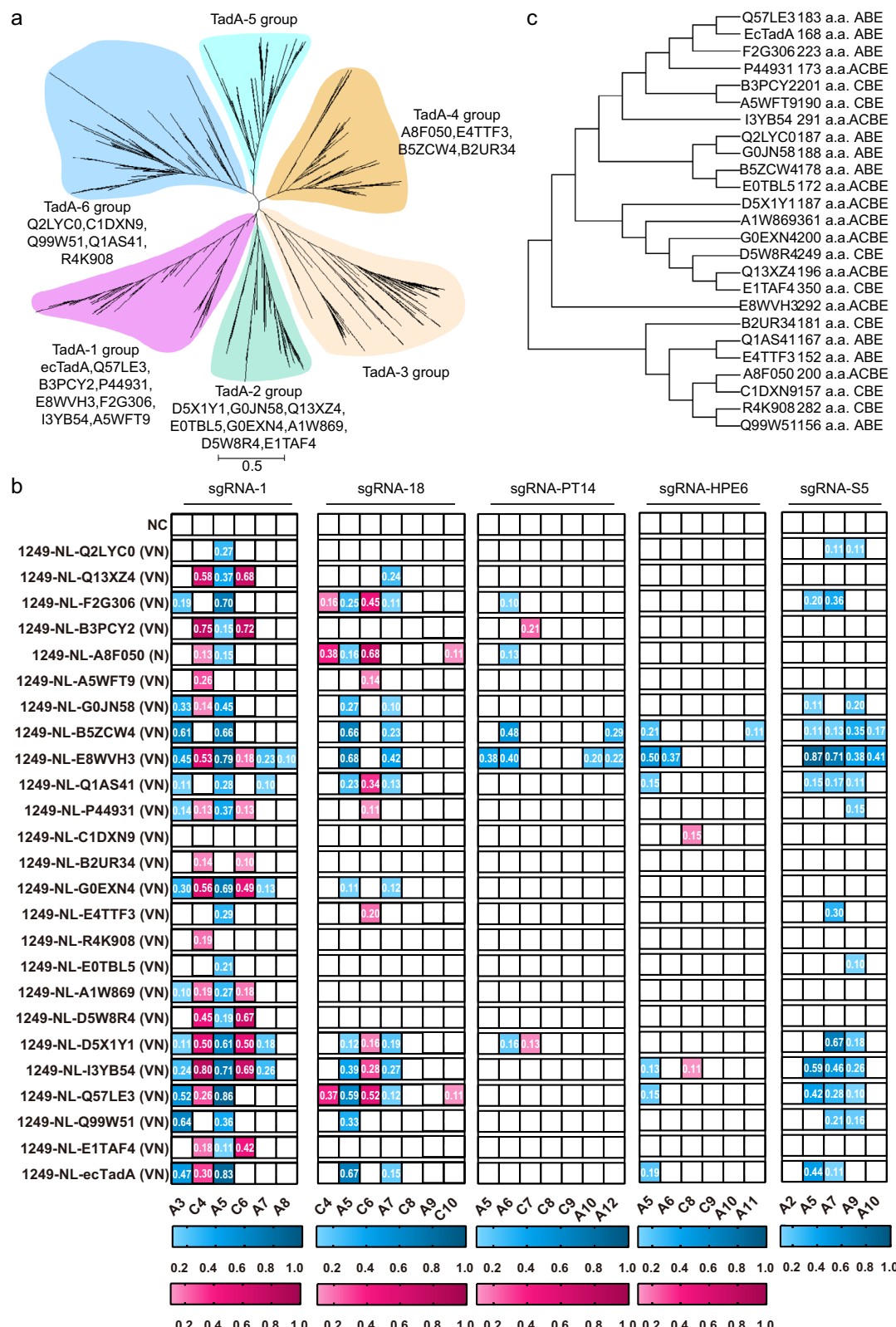

**Fig. 2 | Engineering of TadA orthologs for functional ABEs, CBEs, and ACBEs.**
**a** Phylogenetic analysis of TadA orthologs with a maximum-likelihood tree and six
TadA groups were defined. The evolutionary distance scale of 0.5 was shown.
**b** Editing signatures of selected TadA ortholog-derived base editors at sgRNA-1,
sgRNA-18, sgRNA-PT14, sgRNA-HPE6, and sgRNA-S5 were shown in heatmap for-
mat, with adenine editing efficiencies (A-to-G editing) shown in blue and cytosine
editing efficiencies (C-to-T editing) in pink gradient color. Data shown here

represent means of results from $n = 2$ biologically independent experiments.
**c** Phylogenetic tree showing the relationship between 25 functional TadA ortho-
logs. Protein lengths and substrate specificities of derived base editors were pre-
sented simultaneously. NC negative control. VN, amino acid substitutions
corresponding to A106V&D108N of ecTadA. Source data are provided as a Source
Data file.

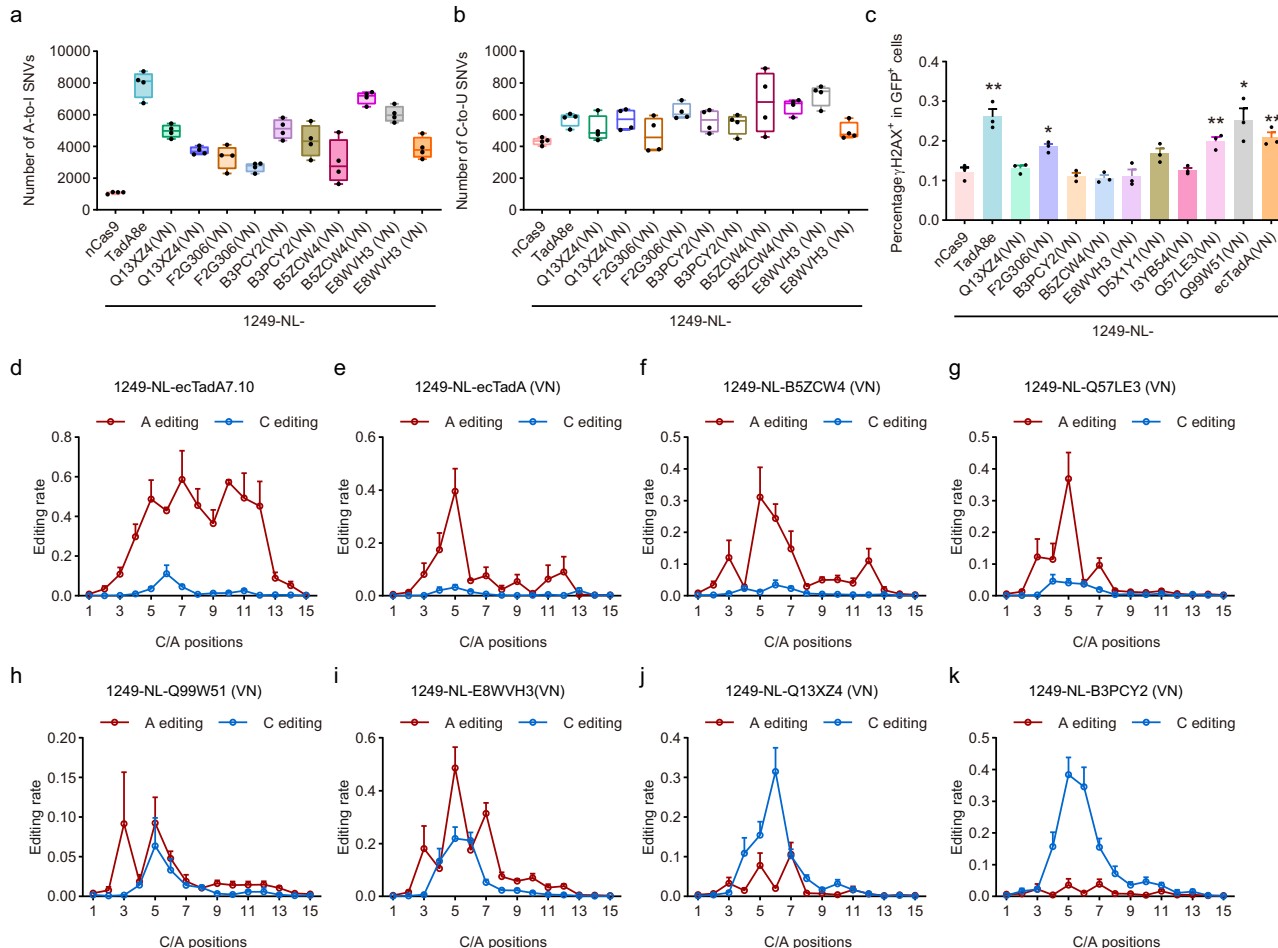

**Fig. 3 | Off-target effects, genotoxicity, and editing scopes of representative TadA ortholog-derived base editors. a, b** Box plots showing the number of RNA A-to-I (**a**) and C-to-U edits (**b**) induced by representative base editors or nCas9 control. $n = 4$ biologically independent experiments. Box plots here are defined by whiskers in terms of minima and maxima, and the center and bounds of the box by quartiles (Q1–Q3). **c** Quantification of γH2AX signaling in HEK293T cells transfected with representative base editors. The percentage of γH2AX positive population within GFP positive cells is shown. Data shown here represent means of results from biologically independent experiments. Data were presented as mean values ± SEM. Exact *P* values were as follows, $P = 0.0037$/TadA-8e, 0.01/F2G308(VN), 0.0084/

Q57LE3(VN), 0.018/Q99W51(VN), and 0.0081/ecTadA(VN), respectively. **d–k** Editing scopes of representative TadA ortholog-derived base editors across 12 endogenous sites, including 1249-NL-ecTadA7.10 (**d**), 1249-NL-ecTadA(VN) (**e**) as control, 1249-NL-B5ZCW4(VN) (**f**), 1249-NL-Q57LE3(VN) (**g**), 1249-NL-Q99W51(VN) (**h**), 1249-NL-E8WVH3(VN) (**i**), 1249-NL-Q13XZ4(VN) (**j**), and 1249-NL-B3PCY2(VN) (**k**). * represents $P < 0.05$, ** represents $P < 0.01$ with two-tailed unpaired *t*-test. Adenine editing (A editing) was shown in red lines and cytosine editing (C editing) was shown in blue lines. Data were presented as mean values ± SEM. Source data are provided as a Source Data file.

conserved residues could generate functional ABEs with obvious adenine editing activity and detectable cytosine editing activity, consistent with previous studies showing that classical ABEs containing evolved ecTadA displayed cytosine editing activity within specific sequence contexts[18,28–31]. By screening tens of different TadA orthologs in HEK293T cells with flow cytometry (Supplementary Fig. 3), we illustrated that two amino acid substitutions combined with an internal fusion strategy could generate functional ABEs, ACBEs, or CBEs with distinct TadA orthologs, providing promising and alternative avenues to expand the toolkits of base editors with improved editing signatures such as minimized Cas9-independent DNA off-target effects for CBEs and smaller size for ACBEs. What's more, their RNA off-target edits could also be minimized or even eliminated through additional mutagenesis. In consideration of potent base editing capacities at more endogenous sites (Fig. 3d–k), reduced DNA and RNA off-target editing effects that could be further minimized (Figs. 3a–c, 4 and Supplementary Fig. 2), orthologs B5ZCW4, Q57LE3, E8WVH3, Q13XZ4, and B3PCY2 were recommended as representative promising candidates for further engineering. We noticed that several other TadA orthologs, such as I3YB54 and D5X1Y1, also displayed robust base

editing capacities in our deaminase screening process (Fig. 2b). Nevertheless, additional editing assessment against more endogenous sites and DNA/RNA off-target analysis are required to provide a comprehensive evaluation of their editing signatures, which would be important for further engineering of these orthologs in the future.

It was noticed that 1249-NL-Q99W51(VN) was initially considered as ABE in our deaminase screening against five endogenous sites (Fig. 2b, c), while the systematic editing evaluation against twelve endogenous sites revealed that it displayed low and comparable adenine and cytosine editing activities at A3, A5, and C5 (Fig. 3h), indicating that 1249-NL-Q99W51(VN) displayed position-specific C-to-T editing capacity. This discrepancy could be attributed to different cytosine positions at endogenous sites analyzed during two processes, as none of the five sites used in deaminase screening contained cytosine at the C5 position, while 12 endogenous sites almost covering cytosines at different positions, which enabled the detection of position-specific CBE activity. Therefore, further analysis of the editing signatures for orthologous TadA-derived base editors described in this study against more sites, such as target library analysis[35,36], would be valuable to reveal potential

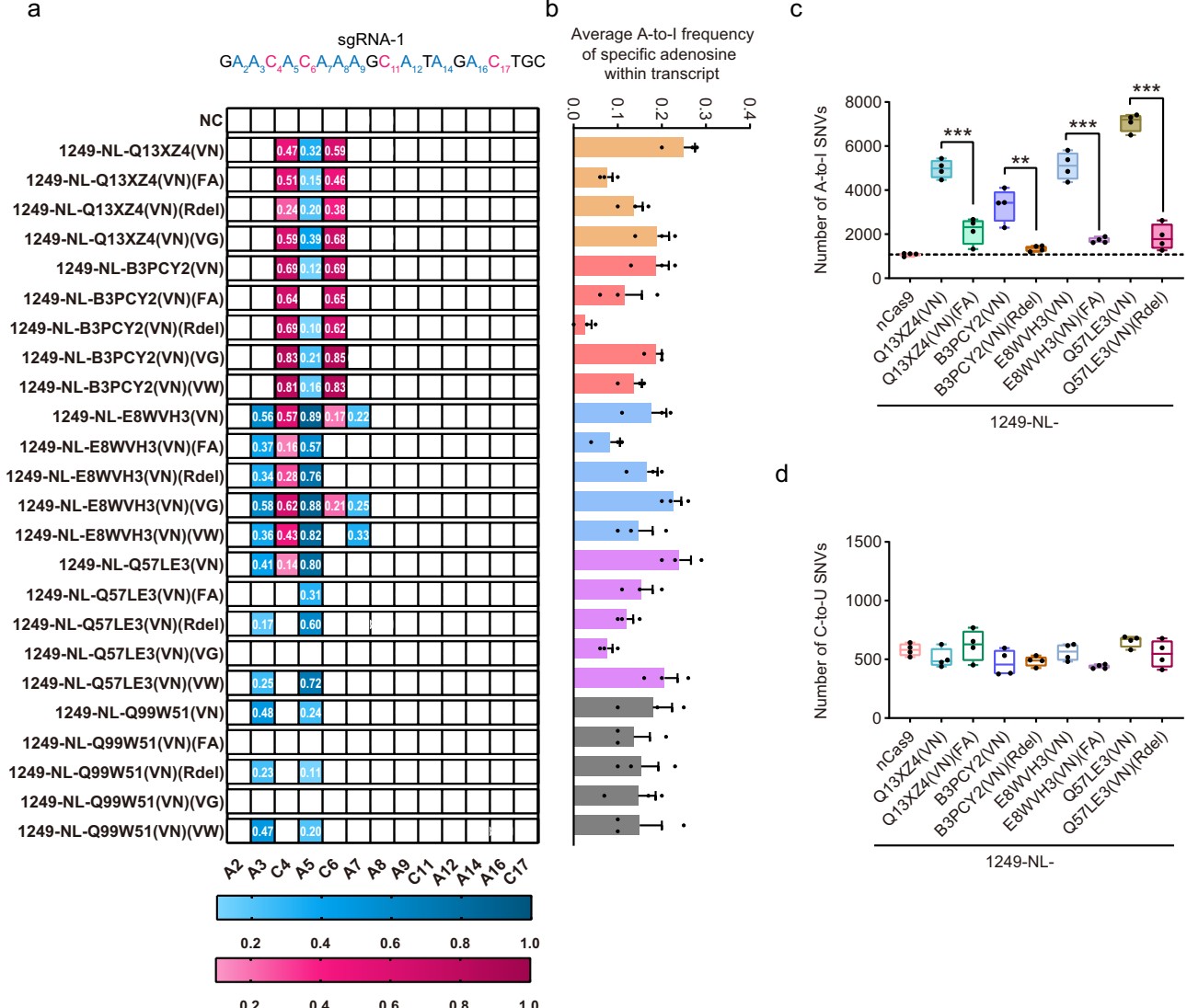

**Fig. 4 | Minimization of RNA off-target effects for TadA ortholog-derived base editors. a, b** Deaminase mutagenesis screening to minimize RNA off-target edits of representative TadA ortholog-derived base editors. Heatmap format showing the on-target editing activity at sgRNA-1, with adenine editing activity in blue and cytosine editing activity in pink (**a**) and histogram showing RNA A-to-I conversion frequency at a specific site within mRNA transcript (**b**). NC, negative control. VN, FA, Rdel, VG, and VW, amino acid substitutions corresponding to A106V&D108N, F148A, R153 deletion, or V82G/W of ecTadA, respectively. RNA A-to-I conversion frequency of base editors containing Q13XZ4, B3PCY2, E8WVH3, Q57LE3, and Q99S1 were depicted in orange, pink, blue, purple, and gray, respectively. Data shown here represent means of results from $n = 3$ biologically independent experiments. Data were presented as mean values ± SEM. **c, d** Box plots showing the number of RNA A-to-I (**c**) and C-to-U edits (**d**) induced by engineered TadA ortholog-derived base editors or nCas9 control. Box plots here are defined by whiskers in terms of minima and maxima, and the center and bounds of the box by quartiles (Q1–Q3). $n = 4$ biologically independent experiments. ** represents $P < 0.01$ and *** represents $P < 0.001$ with two-tailed unpaired $t$-test. Exact $P$ values were as follows, $P = 0.000241$, 0.002087, 4.24e-05, and 6.33e-06, respectively. Source data are provided as a Source Data file.

position-specific or context-specific base editing activities and to guide the development of base editing tools with a specific position or/and context preference.

We demonstrated in another study published simultaneously that ecTadA variant TadA-8e could be reprogrammed to generate potent CBEs and ACBEs with minimized DNA/RNA off-target edits. Additionally, reprogrammed TadA-8e variants enabled the generation of potent miniCBEs with high precision and minimized off-target effects[37]. Considering that the engineering of TadA orthologs enabled the generation of functional ABEs, CBEs, and ACBEs in this study, adenosine deaminases TadAs represented alternative strategies to develop optimized CBEs and miniCBEs. In the future, systematic exploration of TadA orthologs, combined with critical amino acids identified during the TadA-8e reprogramming process, would guide the generation of

conventional or miniature base editors with diversified editing signatures to expand current toolkits.

It is widely believed that cytidine deaminases are required for CBEs while adenosine deaminases with complicated engineering are required for ABEs[1,2]. Nevertheless, cytidine and adenosine deaminases have a similar active pocket and highly conserved catalytic H(C)XE and PCXXC motifs, indicating an evolutionary relationship between them[38]. Recent studies have also demonstrated that evolved TadA variant TadA-8e could be engineered to enable robust C-to-T editing with or without A-to-G editing capacity[30,31], further supporting a potential evolutionary relationship between cytidine and adenosine deaminases. However, a single natural protein possessing both C-U and A-I conversion activities has not been identified, and only one enzymatic

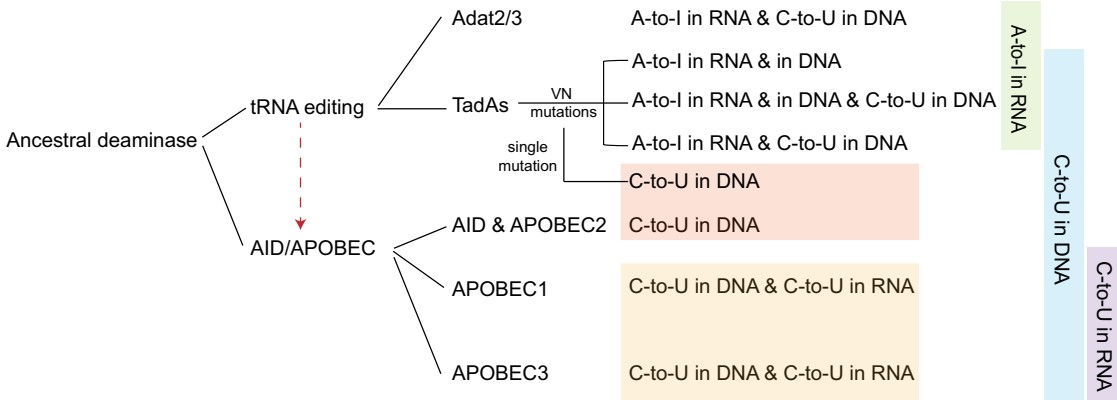

**Fig. 5 | Simplified phylogenetic tree of tRNA-editing deaminases and AID/APOBEC deaminase family.** Substrate specificity of different deaminases was displayed in a phylogenetic tree, with TadAs showing all different substrate specificities of engineered orthologs in this study. DNA-specific C-U deamination was highlighted in orange, while both DNA and RNA-targetable C-to-U deamination was highlighted in yellow. Additionally, the shared substrate specificity of different deaminases was highlighted in the right panel, with A-to-I in RNA in green, C-to-U in DNA in blue, and C-to-U in RNA in purple. The red dash arrow indicated that AID/APOBEC might be derived from tRNA-editing deaminases.

complex, TbADAT2/3 from *Trypanosoma brucei*, has been reported to mediate both C-U deamination in ssDNA and A-I in tRNA in vitro[39]. In consideration of the engineering results of TadA orthologs, we constructed a simplified phylogenetic tree for adenosine and cytidine deaminases with respective deamination capacity against different substrates (Fig. 5) to elucidate the potential evolutionary process of deaminases. It was noticed that TadA orthologs could be engineered with just VN double mutations to display cytosine deamination activity in ssDNA, in addition to the intrinsic RNA adenine deamination activity. Furthermore, the intrinsic RNA adenine deamination activity of TadAs could be further eliminated with a single mutation. For the AID/APOBEC cytidine deaminase family, the ancestral member's AID and APOEC2 just displayed cytosine deamination activity at ssDNA but not RNA level while later derived APOBEC1/3 further obtained RNA cytosine deaminase activity[40]. According to the above analysis, it is possible that tRNA-specific adenosine deaminases might be the ancestral members, with the AID/APOBEC cytidine deaminase family as later arrivals, derived from TadAs. In consistent with this hypothesis, it has been demonstrated that AID/APOBEC deaminases bind substrates with similar conformations to tRNA[40], the substrate of TadAs, further supporting the speculation that cytidine deaminases might evolve from tRNA-editing enzymes[39,41].

In summary, we successfully generated functional ABEs, CBEs, and ACBEs, with reduced or minimized off-target effects, through screening and engineering of TadA orthologs. Our results not only expand the toolkits of base editing, but also provide important evidence for the evolution of editing deaminases.

## Methods
### Plasmid construction
TadA-8e and codon-optimized engineered orthologous TadAs were synthesized commercially (Synbio Technologies). The mutagenesis strategy (KOD-plus, Toyobo, Cat#: KOD-201) was used to induce specific mutations for corresponding base editors. To generate TadA ortholog-derived base editors, the docking site at residue 1249 was initially inserted with SpeI-NLS-BamHI-NheI-linker-XbaI sequence, and then engineered orthologous TadAs were inserted via BamHI-NheI double digestion strategy. Plasmids expressing dSaCas9-UGI-T2A-mCherry and U6-sgsaRNA, which were used in R-loop assay, were derived from PX602 (Addgene #61593), in which D10A and N580A were induced via mutagenesis strategy (KOD-plus, Toyobo, Cat#: KOD-201), and then UGI-T2A-mCherry cassette was inserted through BamHI-EcoRI double digestion strategy. SgRNA expression vectors were U6-sgRNA-EF1alpha-UGI-T2A-mCherry as described previously, and to generate plasmid expressing sgRNAs for puromycin-based enrichment, mCherry was replaced with a puromycin resistance gene. Information for sgRNAs used in this study was listed in Supplementary Data 4.

### Cell transfection
Dulbecco's modified Eagle's medium (DMEM, Sigma-Aldrich; Cat#: D5796) supplemented with 10% FBS (Gibco, Thermo Fisher Scientific; Cat#: 26010074) were used for HEK293T cells (GNHu17, Cell Bank of the Chinese Academy of Sciences, Shanghai, China) and cells were cultured at 37 °C in 5% $CO_2$ incubator (Heraeus, Thermo Fisher Scientific). Twenty-four hours before transfection, HEK293T cells were passaged and plated into six-well or 48-well plates (Corning). Base editor- and sgRNA-expressing plasmids (mole ratio 2:1) were mixed with 2.5 µl (48-well) or 8 µl (six-well) Lipo293™ (Beyotime Biotechnology, Shanghai, China; Cat#: C0521) for transient transfection. For base editing analysis, cells were plated and transfected in 48-well plates and cultured for another 72 h. Then Flow cytometry (Moflo XDP, Beckman Coulter/BD FACSAria™ Fusion Flow Cytometers) was performed to collect double-positive cells. For transcriptome analysis, cells were plated and transfected in six-well plates. Twenty-four hours later, the medium were replaced with fresh DMEM containing 2 µg/ml puromycin (Beyotime Biotechnology, Shanghai, China; Cat#: ST551-10mg) and cultured for another 48 h, then total RNA were collected for high-throughput RNA sequencing.

### Editing analysis
Double-positive cells collected as described above were treated with DirectPCR reagent (Viagene Biotech, Ningbo, China; Cat#: 302-C). Targeted amplifications were produced using LA Taq (Takara, Dalian, China; Cat#: RR02MA). Sanger sequencing results were quantified with EditR software and targeted amplicon sequencing results were analyzed with CRISPResso2 (Shanghai Personalbio Technology, Shanghai, China). Targeted amplicon sequencing was performed using the Illumina NovaSeq platform at Shanghai Personalbio Technology. Experiments for amplicon sequencing were performed in biological triplicates.

### Phylogeny comparison of TadA orthologs
Most TadA orthologs used for phylogenetic tree analysis were described previously[32], with the addition of several TadAs having

reported structures in PDB (TadA orthologs used for phylogenetic tree analysis were listed in Supplementary Data 1). Multiple sequence alignment was performed using MAFFT, and a maximum-likelihood phylogenetic tree was constructed using MEGA.

## Screening and engineering of TadA orthologs

To select representative TadA orthologs, TadA orthologs were initially aligned with BLOSUM62 using Jalview2.11.2.5. Selection criteria include (1) >140 a.a. length to exclude potential partial sequences (16 orthologs were excluded); (2) structural information (four included, *Escherichia coli* TadA (ecTadA), *Staphylococcus aureus* TadA (Q99W51), *Salmonella choleraesuis* TadA (Q57LE3) and *Streptococcus pyogenes* TadA (Q5XE14)); (3) evolution distances, 50 TadA orthologs with large evolution distances were included to ensure the high diversity. To design TadA orthologs for functional screening, conserved residues corresponding to A106 and D108 residues of ecTadA were substituted with valine (V) and asparagine (N), respectively, as A106V and D108N are core mutations for ecTadA to be adapted for functional ABE. Overall, six TadA orthologs already have N substitution at the residue corresponding to D108 and only A-V mutation was induced at the residue corresponding to A106 of ecTadA, while one TadA ortholog already have V substitution at the residue corresponding to D108 and only D-N mutation was induced at the residue corresponding to D108 of ecTadA. Additionally, four TadA orthologs have no obvious conserved residues and thus, no mutations were induced. Selected TadA orthologs were referred to as UniProt ID with mutations corresponding to A106 and D108, such as V, N, or VN.

## R-loop assay for Cas9-independent DNA off-target analysis

Generally, base editor-expressing plasmid was co-transfected into 48-well HEK293T cells with particular dSaCas9-UGI-T2A-mcherry-U6-sgsaRNA and cultured for 72 h. Double-positive cells were collected by flow cytometry for targeted amplicon sequencing.

## RNA editing quantification

Total RNA was extracted using Trizol reagent (Life Technologies, Thermo Fisher Scientific) for RNA sequencing. Illumina Hiseq (PE 2◊150), at a depth of ~20 million reads per sample was used. The long-RNA-seq-pipeline of ENCODE Consortium was used for RNA sequencing analysis, with GRCh38 reference genome using annotation GRCh38.v96 via STAR (v2.4.2a) in 2-pass mode used for alignment[42]. Variant calling was performed with Sentieon® genomics tools (v202010.02). After removing duplicates with Picard (v2.23.6, http://broadinstitute.github.io/picard), reads were split at junctions into exon segments and reassigned the mapping qualities from STAR. Base-quality score recalibration (BQSR) was performed as the DNA-seq to remove potential experimental biases. MuTect2 (202010.02) was then used to identify variants, and the VariantFiltration tool (gatk-4.1.4.0) with parameter --filter-expression "QUAL <25|| MQ <20.0|| QD <2.0|| FS >30.0|| DP <20" was used to filter variants with base-quality score <25, mapping quality score <20, Fisher strand values >30.0, qual by depth values <2.0 or sequencing depth <20. Variants in GFP-only expressing cells were considered as SNPs and filtered out from other groups.

## H2AX staining

For H2AX staining, cells were collected and washed with PBS, and fixed with 4% paraformaldehyde at room temperature (25 °C) for 15 min. Following washes with PBS, cells were permeabilized with 0.5% Triton X-100 in PBS for 2 min. Next, block cells with buffer (10% goat serum in TBS) at room temperature (25 °C) for 1 h. For the detection of DNA damage, stain the cells with γ-H2AX antibody (Cat#: 560447, BD Pharmingen, 1:10 at dilution) at 4 °C for 2 h and wash with PBS. Cells were gated on fluorescein isothiocyanate and allophycocyanin using the Fortessa Flow Cytometer (BD Biosciences), and results were analyzed by FlowJo v10.

## Statistics and reproducibility

Statistical analyses were performed using Prism 8.4.0 (GraphPad). The results are presented as mean (heatmap) or mean ± standard error of the mean (SEM) (bar plot). Two-tailed unpaired *t*-test was used for two group comparisons in Figs. 3, 4. $P < 0.05$ was considered as statistical significance, with * represents $P < 0.05$, ** represents $P < 0.01$, and *** represents $P < 0.001$. RNA-seq analysis was conducted using STAR (v2.4.2a), Sentieon® genomics tools (v202010.02), Picard (v2.23.6), MuTect2 (202010.02), and VariantFiltration tool (gatk-4.1.4.0). $N = 2$ to 4 independent biological replicates were performed and demonstrated in each figure. In this study, no statistical method was used to predetermine sample size, and no data were excluded from the analyses. The experiments were not randomized and the Investigators were not blinded to allocation during experiments and outcome assessment.

## Reporting summary

Further information on research design is available in the Nature Portfolio Reporting Summary linked to this article.

## Data availability

The raw high-throughput sequencing data generated in this study have been deposited in the NCBI sequence Read Archive database under PRJNA758206 and PRJNA757902. GRCh38 reference genome GRCh38.v96 used in this study is available at https://ftp.ensembl.org/pub/release-96/gtf/homo_sapiens/Homo_sapiens.GRCh38.96.gtf.gz. All plasmids described in this work are available and please contact the corresponding author, T.-L.C. (chengtianlin@fudan.edu.cn) and will be deposited to Addgene. Source data are provided with this paper.

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

## Acknowledgements

We thank Yuefang Zhang and Shifang Shan for their help in cell culture experiments. Haiyan Wu and Lijuan Quan for their help in flow cytometry. This work was supported by grants from the National Key R&D Program of China (2019YFA0111000), Natural Science Foundation of Shanghai (20ZR1403100), Shanghai Municipal Science and Technology (20JC1419500) to T.-L.C., Supported by the Lingang Laboratory (Grant No. LG-QS-202203-08) to T.-L.C., National Natural Science Foundation of China (#31600826) to T.-L.C. National Key R&D Program of China (2020YFA0712403), National Natural Science Foundation of China (T2225015 and 61932008), Shanghai Municipal Science and Technology Major Project (2018HZDZX01) and ZJLab to X.-M.Z. National Natural Science Foundation of China (#32000726) to B.Y. The research is supported by the Open Large Infrastructure Research of the Chinese Academy of Sciences.

## Author contributions

T.-L.C. designed the study. S.Z., J.L.C., and J.Q. conducted the experiments. L.S., B.Y., and J.X.C. performed computational analysis. T.-L.C., J.Q.C., X.-M.Z., and Z.Q. supervised the study. T.-L.C. wrote the manuscript with the approval of all other authors.

## Competing interests

Fudan University has a patent (Chinese Patent Application No. 202111226226.X) pending, with T.-L.C., S.Z., and J.Y.Q. as inventors, for TadA orthologs described in this paper. The remaining authors declare no competing interests.
