## [Peer Review File · Nature Communications]

Reviewers' Comments:

Reviewer #1:

Remarks to the Author:

In this manuscript, Zhang et al. have identified 25 TadA orthologs by selection from total 1000 ones and demonstrated that they could be engineered to generate functional ABEs, CBEs or ACBEs by previously described engineered strategy. Overall, the novelty of this study lie in that use of one deaminase could concurrently produce C to U and A to I mutation, which be achieved by incorporating two or more deaminases previously. However, the whole work seems poorly designed, and many places were not clearly presented. The main issues are as follows.

- 1.The authors described they obtained 1000 TadA orthologs for phylogenetic tree analysis and classified 61 into 6 groups. Could you please incorporate more details at least in Method part about how you get 1000 TadA orthologs or perform the phylogenetic tree?
- 2.Similarly, how the 54 TadA orthologs were chosen from 1000 ones? Are there any selection criteria?
- 3.About the experiment design for ABE, since the ABE7.10 was structurally constructed by putting ecTadA on N terminus of Cas9, why you directly insert TadA orthologs into residue 1249 site of nCas9 without any explanation? Although it was reported that deaminase imbedding could reduce off-target, the efficiency varied from different insertion sites. In fact, as Figure 1, the 1249 residue insertion greatly reduce the editing efficiency of ecTadA.
- 4.In figure 1, what's the "NL" in naming each ABEs? I strongly recommend illustrate the structure of ABE construct.
- 5.In figure 2d, it is better by incorporating ecTadA-derived ABE as comparison.
- 6.In figure 2e, in the right histogram, how do you define A-to-I editing frequency?
- 7.The authors described their study providing important evidence for the evolution of editing deaminase. What is evidence? Could you please present more details?
- 8.The author should at least add the discussion part especially describing how the work advance the current understanding of BEs.

Reviewer #2:

Remarks to the Author:

* General comments:

Base editors including cytosine base editors (CBEs) and adenine base editors (ABEs) are innovative genome editing tools capable of precise base conversion without donor DNA and causing DNA double strand breaks. However, the gRNA-independent genome-wide DNA deamination effect in particular of CBEs and/or transcriptome-wide RNA deamination would hinder further therapeutic applications. In this manuscript, the authors aimed to develop alternative CBE, ABE, ACBE system by adopting other TadA enzyme orthologs, not E. coli driven TadA. The authors found 25 TadA orthologs potentially capable of ABEs, CBEs, or ACBEs via single/double mutations and ultimately found optimized base editors with minimized Cas9-independent DNA off-target effects and genotoxicity. Overall, this study is straightforward and the topic would be interesting for broad readers in the genome editing field, thus is potentially suitable for this journal, Nature Communications. I would like to raise several issues to strengthen this study.

* Specific comments:

1. Overall, the main text is too short and detailed information lack. For example, in this study, TadA enzymes were inserted into the residue 1249 cite of nCas9 to generate base editors without detailed description. In the previous paper [ref #18], the same group suggested docking sites of nCas9 for better specificity. But the detailed description why the authors used this approach, (not fused to N/C-terminus of nCas9), is required for broad readers.

2. As shown in Figure 1b, the authors tested tens of TadA orthologs, but it is hard to directly know the base editing activities of them. The absolute values of editing efficiencies, rather than the relative ratios with heatmap, should be provided.

2.1. In addition, as a positive control, editing efficiencies of ecTadA7.10 or TadA8e should be provided together.

3. It is a little ambiguous to know the name of each ortholog. It is necessary to explain the nomenclature of each ortholog.

4. As a result, what types do the authors recommend for ABE, CBE, and ACBE? It is better to suggest them in the abstract.

(Minor)

- In line 101, "are representative cases." should be removed.
- "1249 site" are sometimes noted as "49" including lines 102-113.

REVIEWER COMMENTS

Reviewer #1 (Remarks to the Author):

In this manuscript, Zhang et al. have identified 25 TadA orthologs by selection from total 1000 ones and demonstrated that they could be engineered to generate functional ABEs, CBEs or ACBEs by previously described engineered strategy. Overall, the novelty of this study lie in that use of one deaminase could concurrently produce C to U and A to I mutation, which be achieved by incorporating two or more deaminases previously. However, the whole work seems poorly designed, and many places were not clearly presented. The main issues are as follows.

1. The authors described they obtained 1000 TadA orthologs for phylogenetic tree analysis and classified 61 into 6 groups. Could you please incorporate more details at least in Method part about how you get 1000 TadA orthologs or perform the phylogenetic tree?

Response: We thank the reviewer's comments. More details were provided for phylogenetic tree analysis as follows in The method part "Phylogeny comparison of TadA orthologs. Most TadA orthologs used for phylogenetic tree analysis were described previously³⁰, with addition of several TadAs having reported structures in PDB (TadA orthologs used for phylogenetic tree analysis were listed in Supplementary Table 1). Multiple sequence alignment was performed using MAFFT, and maximum-likelihood phylogenetic tree was constructed using MEGA."

2. Similarly, how the 54 TadA orthologs were chosen from 1000 ones? Are there any selection criteria?

Response: We thank the reviewer's comments. More details were provided in the method part as follows: "To select representative TadA orthologs, TadA orthologs were initially aligned with BLOSUM62 using Jalview2.11.2.5. Selection criteria includes (1) >140 a.a. length to exclude potential partial sequences (16 orthologs were excluded); (2) structural information (4 included, *Escherichia coli* TadA (ecTadA), *Staphylococcus aureus* TadA (Q99W51), *Salmonella choleraesuis* TadA (Q57LE3) and *Streptococcus pyogenes* TadA (Q5XE14)); (3) evolution distances, 50 TadA orthologs with large evolution distances were included to ensure the high diversity."

Additionally, we also revised the result part as follows: "54 TadA orthologs with large evolution distances across all 6 groups, including four orthologs with known structural information (*Escherichia coli* TadA (ecTadA), *Staphylococcus aureus* TadA (Q99W51), *Salmonella choleraesuis* TadA (Q57LE3) and *Streptococcus pyogenes* TadA (Q5XE14))".

3. About the experiment design for ABE, since the ABE7.10 was structurally constructed by putting ecTadA on N terminus of Cas9, why you directly insert TadA orthologs into residue 1249 site of nCas9 without any explanation? Although it was reported that deaminase imbedding could reduce off-target, the efficiency varied from different insertion sites. In fact, as Figure 1, the 1249 residue insertion greatly reduce the editing efficiency of ecTadA.

Response: We thank the reviewer's suggestions. We included detailed information to describe why we chose residue 1249 site of nCas9 to generate base editors as figure 1 in the revised manuscript. Briefly, we evaluated the editing capacity of ecTadA(VN) (wild-type *E. coli* TadA carrying only

A106V and D108N double mutations) fusing at N- or representative internal insertion sites of nCas9, and revealed that 1249-NL-ecTadA(VN) displayed optimal adenine and cytosine editing capacities. Therefore, DS1249 insertion site and modified deaminase domain containing 5'-NLS and 3'-flexible linker (hereafter NL-deaminase) were selected for subsequent experiments.

Additionally, For the comment “In fact, as Figure 1, the 1249 residue insertion greatly reduce the editing efficiency of ecTadA.”, we apologize that we did not describe the data in original manuscript clearly. Actually, in Figure 1 of original manuscript, ecTadA(VN) represented wild-type *E.coli* TadA carrying only A106V and D108N double mutations. In our revised manuscript, we demonstrated that 1249-NL-ecTadA(VN) displayed optimal adenine and cytosine editing capacities as compared to N- or other internal fusing sites (Fig. 1 in revised manuscript).

4.In figure 1, what's the “NL” in naming each ABEs? I strongly recommend illustrate the structure of ABE construct.

Response: We thank the reviewer’s suggestions. “NL” means 5'-NLS and 3'-flexible linker around deaminase domain. In the revised manuscript, we provided structure of ABE construct in Figure 1a and 1c accordingly.

5.In figure 2d, it is better by incorporating ecTadA-derived ABE as comparison.

Response: We thank the reviewer’s suggestions. In the revised manuscript, we included ecTadA(VN)- (1249-NL-ecTadA(VN)) and ecTadA7.10-derived ABEs (1249-NL-ecTadA7.10) as comparison.

6.In figure 2e, in the right histogram, how do you define A-to-I editing frequency?

Response: In the right histogram of original figure 2e, A-to-I editing frequency means the A-to-I conversion frequency of specific adenosine within mRNA transcript. In brief, we extract total RNA from cells transfecting corresponding ABEs for reverse transcription. Then targeted amplification was performed using cDNA to examine the A-to-I frequency of specific adenosine within mRNA transcript.

7.The authors described their study providing important evidence for the evolution of editing deaminase. What is evidence? Could you please present more details?

Response: We thank the reviewer’s comments and suggestions. In the revised manuscript, we added Fig.5 to elucidate the possible evolutionary process of adenosine and cytidine deaminases. We proposed that TadAs might be the ancestral members and AID/APOBEC family might be derived from TadAs, as TadA engineering with two amino acid substitutions generated functional CBEs, and intrinsic RNA A-to-I deamination activity could be eliminated by additional single mutation. The detailed discussion was included in the revised manuscript.

8.The author should at least add the discussion part especially describing how the work advance the current understanding of BEs.

Response: We thank the reviewer’s suggestions. In the revised manuscript, we added the discussion part accordingly and demonstrated that our work provided critical insights into BE development. Briefly, our work demonstrated that (1) in combined with internal fusion strategy, TadA deaminases could be engineered simply via one or two amino acid substitutions to generate functional ABEs.

(2) provided a simplified strategy by one or two amino acid substitutions in combined with internal fusion for TadA screening to identify potential TadA orthologs suitable for the generation of functional ABEs. (3) generated diversified ABEs, ACBEs and CBEs using distinct TadA orthologs to expand the toolkits of base editors with improved editing signatures.

Reviewer #2 (Remarks to the Author):

** General comments:*

Base editors including cytosine base editors (CBEs) and adenine base editors (ABEs) are innovative genome editing tools capable of precise base conversion without donor DNA and causing DNA double strand breaks. However, the gRNA-independent genome-wide DNA deamination effect in particular of CBEs and/or transcriptome-wide RNA deamination would hinder further therapeutic applications. In this manuscript, the authors aimed to develop alternative CBE, ABE, ACBE system by adopting other TadA enzyme orthologs, not E. coli driven TadA. The authors found 25 TadA orthologs potentially capable of ABEs, CBEs, or ACBEs via single/double mutations and ultimately found optimized base editors with minimized Cas9-independent DNA off-target effects and genotoxicity. Overall, this study is straightforward and the topic would be interesting for broad readers in the genome editing field, thus is potentially suitable for this journal, Nature Communications. I would like to raise several issues to strengthen this study.

** Specific comments:*

1. Overall, the main text is too short and detailed information lack. For example, in this study, TadA enzymes were inserted into the residue 1249 cite of nCas9 to generate base editors without detailed description. In the previous paper [ref #18], the same group suggested docking sites of nCas9 for better specificity. But the detailed description why the authors used this approach, (not fused to N/C-terminus of nCas9), is required for broad readers.

Response: We thank the reviewer's suggestions. We included detailed information to describe why we chose residue 1249 site of nCas9 to generate base editors as figure 1 in the revised manuscript. Briefly, we evaluated the editing capacity of ecTadA(VN) (wild-type *E.coli* TadA carrying only A106V and D108N double mutations) fusing at N- or representative internal insertion sites of nCas9, and revealed that 1249-NL-ecTadA(VN) displayed optimal adenine and cytosine editing capacities. Therefore, DS1249 insertion site and modified deaminase domain containing 5'-NLS and 3'-flexible linker (hereafter NL-deaminase) were selected for subsequent experiments.

2. As shown in Figure 1b, the authors tested tens of TadA orthologs, but it is hard to directly know the base editing activities of them. The absolute values of editing efficiencies, rather than the relative ratios with heatmap, should be provided.

Response: We thank the reviewer's suggestions. We have revised the manuscript accordingly to display the absolute values of editing efficiencies.

2.1. In addition, as a positive control, editing efficiencies of ecTadA7.10 or TadA8e should be provided together.

Response: We thank the reviewer's suggestions. In the revised manuscript, we included ecTadA(VN)- (1249-NL-ecTadA(VN)) and ecTadA7.10-derived ABEs (1249-NL-ecTadA7.10) as comparison.

3. It is a little ambiguous to know the name of each ortholog. It is necessary to explain the nomenclature of each ortholog.

Response: We thank the reviewer's comment. As a large amount of TadA orthologs were analyzed in our manuscript, we chose Uniprot ID for orthologs nomenclature.

4. As a result, what types do the authors recommend for ABE, CBE, and ACBE? It is better to suggest them in the abstract.

Response: We thank the reviewer's suggestions. In the revised manuscript, we suggested that "with orthologs B5ZCW4, Q57LE3, E8WVH3, Q13XZ4 and B3PCY2 as promising candidates for further engineering." in the abstract.

(Minor)

- In line 101, "are representative cases." should be removed.

Response: We thank the reviewer's suggestions, and we have revised the manuscript accordingly.

- "1249 site" are sometimes noted as "49" including lines 102-113.

Response: We thank the reviewer's suggestions. We have checked and revised the manuscript accordingly.

Reviewers' Comments:

Reviewer #1:

Remarks to the Author:

The authors have addressed the majority of my concerns and the current revised version is adequately improved. I only have three minor points.

1. Among six represented TadA orthologs, the Q99W51-derived base editor was shown to act as ABE (figure 2b and c), however, in figure 3h, 1249-NL-Q99W51 (VN) was shown to edit A and C with nearly comparable efficiency at position 5-7. How do you explain this?
2. In the revised manuscript, why do the authors recommend such five TadA orthologs? I saw other ones such as I3YB54, D5X1Y1 show higher editing capacity.
3. Very recently two independent groups have reported engineering of ecTadA variants TadA-8e for both cytosine and adenine base editing (Chen et al., Nat Biotech, 2022; Neugebauer et al., Nat Biotech, 2022), the author should at least cite these two papers in the appropriate place of manuscript.

Reviewer #2:

Remarks to the Author:

The authors have mostly answered the issues I raised in the earlier review. I recommend the publication of this revised version.

REVIEWER COMMENTS

Reviewer #1 (Remarks to the Author):

The authors have addressed the majority of my concerns and the current revised version is adequately improved. I only have three minor points.

1. Among six represented TadA orthologs, the Q99W51-derived base editor was shown to act as ABE (figure 2b and c), however, in figure 3h, 1249-NL-Q99W51 (VN) was shown to edit A and C with nearly comparable efficiency at position 5-7. How do you explain this?

Response: We thank the reviewer's comments. We explained these results in discussion part in the revised manuscript (Line 227-240) to clarify this discrepancy and confusion. In brief, as shown in Fig.3h, 1249-NL-Q99W51 (VN) displayed position-specific C-to-T editing capacity, mainly targeting C5, which could be only uncovered by editing analysis at more endogenous sites containing cytosines at almost all different positions. In Fig. 2b and c, the editing activity of 1249-NL-Q99W51 (VN) was just evaluated against five endogenous sites, and none contained cytosine at C5 position while in Fig. 3h, 1249-NL-Q99W51 (VN) was assessed more systematically against 12 endogenous sites almost covering cytosines at different positions, which enabled the detection of position-specific CBE activity.

2. In the revised manuscript, why do the authors recommend such five TadA orthologs? I saw other ones such as I3YB54, D5X1Y1 show higher editing capacity.

Response: We thank the reviewer's comments. We elucidated our recommendation in discussion part in the revised manuscript (Line 217-226). In brief, base editors derived from the five recommended TadA orthologs B5ZCW4, Q57LE3, E8WVH3, Q13XZ4 and B3PCY2 have been characterized comprehensively and displayed improved editing signatures, including potent base editing capacities at more endogenous sites (Fig. 3d-k) and reduced DNA and RNA off-target editing effects that could be further minimized. Base editors generated with I3YB54, D5X1Y1 were only assessed during initial screening analysis at five endogenous sites in Fig. 2, and lack of comprehensive assessment for editing signatures.

3. Very recently two independent groups have reported engineering of ecTadA variants TadA-8e for both cytosine and adenine base editing (Chen et al., Nat Biotech, 2022; Neugebauer et al., Nat Biotech, 2022), the author should at least cite these two papers in the appropriate place of manuscript.

Response: We thank the reviewer's comments. We have cited these two papers in the revised manuscript (Line 63, Line 210). We also cited these two papers to support potential evolutionary relationship between cytidine and adenosine deaminases (Line 245-248).

Reviewer #2 (Remarks to the Author):

The authors have mostly answered the issues I raised in the earlier review. I recommend the publication of this revised version.